# Effects of Oral Exposure to Low-Dose Bisphenol S on Allergic Asthma in Mice

**DOI:** 10.3390/ijms231810790

**Published:** 2022-09-15

**Authors:** Rie Yanagisawa, Eiko Koike, Tin-Tin Win-Shwe, Hirohisa Takano

**Affiliations:** 1Health and Environmental Risk Division, National Institute for Environmental Studies, 16-2 Onogawa, Tsukuba 305-8506, Japan; 2Graduate School of Global Environmental Studies, Kyoto University, Kyoto Daigaku-Katsura, Nishikyo-ku, Kyoto 615-8530, Japan

**Keywords:** bisphenol S, allergic asthma, environmental chemical, Th2 response, endocrine disruptor

## Abstract

Bisphenol S (BPS) is increasingly being used as an alternative for bisphenol A; however, its health effects remain unclear. We investigated the effects of oral exposure to low-dose BPS on allergic asthma. C3H/HeJ male mice were intratracheally administered with allergen (ovalbumin (OVA), 1 μg/animal) every 2 weeks from 6 to 11 weeks old. BPS was ingested by drinking water at doses equivalent to 0.04, 0.4, and 4 μg/kg/day. We then examined pulmonary inflammation, airway hyperresponsiveness, serum OVA-specific immunoglobulin (Ig) levels, Th2 cytokine/chemokine production, and mediastinal lymph node (MLN) cell activities. Compared with OVA alone, moderate-dose BPS (BPS-M) with OVA significantly enhanced pulmonary inflammation, airway hyperresponsiveness, and OVA-specific IgE and IgG1. Furthermore, interleukin (IL)-5, IL-13, IL-33, and CCL11/Eotaxin protein levels in the lungs increased. Conversely, these allergic responses were reduced in the high-dose BPS+OVA group. In MLN cells, BPS-M with OVA increased the total cell count and activated antigen-presenting cells including conventional dendritic cell subset (cDC2). After OVA restimulation, cell proliferation and Th2 cytokine production (IL-4, IL-5, and IL-13) in the culture supernatant also increased. Therefore, oral exposure to low-dose BPS may exacerbate allergic asthmatic responses by enhancing Th2-polarized responses and activating the MLN cells.

## 1. Introduction

Bisphenols are compounds with two hydroxyphenyl groups and are ubiquitously used in polycarbonate plastic and epoxy resin production [1]. Bisphenol A (BPA) is their most representative prototype, commonly found in canned food and beverage linings, toys, dental sealants, and other consumer products [2,3]. Thus, BPA can be found in homes and in the environment, including the air, water, soil, foodstuffs, and house dust [4]. BPA has also been detected in human samples, such as urine [5], blood, breast milk, amniotic fluid, and saliva [6]. Unfortunately, BPA is an endocrine-disrupting chemical (EDC) that can modify the activity of estrogen receptors (ERs), causing adverse effects to human health [7,8,9,10,11]. Consequently, BPA use in manufacturing baby bottles has been banned in some health organizations worldwide, such as the US Food and Drug Administration, European Commission, and Health Canada [12], and BPA alternatives have increasingly been developed.

Bisphenol S (BPS, Figure 1) is one of the most widely used bisphenols. Compared with BPA, BPS has thermal and light stability; therefore, it has been introduced in various industries as an important substitute for BPA [13]. BPS is used as a reagent for polymer reactions and an anticorrosive agent for epoxy adhesives, and it has been included in paper products [14], personal care products [15,16], canned foodstuffs [17], and food [18]. Over the last decade, BPS use has increased significantly, thereby leading to higher environmental exposure to BPS-containing compounds [19,20].

Moreover, BPS shows a longer half-life than BPA and is more resistant to biodegradation [4]. Therefore, BPS is potentially highly persistent and accumulative in the environment. Recently, BPS has been detectable in different environmental matrices, including paper products, surface water, sewage sediments, sludge, indoor dust, and even some foods [16,21,22]. Thermal receipt paper is also potential for occupational exposure to BPS [23]. In addition, the estrogenic activity of BPS is reportedly similar to that of BPA [20,24]; however, the molecular mechanisms attributed to BPS remain poorly understood.

BPA has been associated with asthma and allergic diseases in both animal and human studies [25,26,27,28]. BPA can impact respiratory function and cause airway inflammation in animals [26,27,28]. We also reported that the oral intake of low-dose BPA at levels equivalent to human exposure can aggravate allergic asthmatic responses in mice [29]. A recent epidemiological study reported that urinary BPS was associated with increased odds of existing asthma in male adults [30]. Meanwhile, Quirós-Alcalá showed that BPS and bisphenol F, but not BPA, have no association with asthma symptoms or healthcare utilization [31]. In utero exposure to BPS was not also associated with respiratory outcomes such as current asthma and wheeze [32]. Thus, the relationship between BPS and asthma remains controversial.

In this study, we evaluated the effects of oral exposure to low doses of BPS in a murine model of allergic asthma. We also sought to investigate its effects on mediastinal lymph nodes (MLNs).

## 2. Results

### 2.1. BPS Enhances Allergic Pulmonary Inflammation and Goblet Cell Hyperplasia

To evaluate the effects of BPS oral exposure on allergen-induced pulmonary inflammation, we examined the cellular profile of BAL fluid 48 h after the last intratracheal instillation. The BPS-exposed groups showed no significant changes (Figure 2a). OVA+BPS-M enhanced eosinophil infiltration compared with OVA alone (*p* < 0.05, Figure 2b). Conversely, the OVA+BPS-L and OVA+BPS-H groups exhibited no significant changes. In pulmonary inflammation evaluation using H&E (Figure 3a) and PAS staining, groups without OVA sensitization showed no changes. OVA enhanced eosinophil and lymphocyte accumulation in the peribronchial and perivascular regions and goblet cell hyperplasia in the bronchial epithelium. The accumulation of eosinophils and lymphocytes was more remarkable in the OVA+BPS-M group than in the OVA group (*p* < 0.05, Figure 3b). In the OVA+BPS-H group, pulmonary inflammation in the lungs was rather attenuated. However, goblet cell hyperplasia did not significantly change in either group with OVA.

### 2.2. BPS Promotes Allergen-Induced Airway Hyperresponsiveness

To assess the effects of BPS exposure on airway hyperresponsiveness development in allergic asthmatic mice, we measured airway responsiveness to aerosolized MCh using WBP 24 h after the last OVA instillation. In OVA-sensitized mice, airway responsiveness (percent increase in Penh > PBS baseline) to MCh significantly increased following exposure to BPS-M compared with PBS exposure (Figure 4a). In contrast, the OVA+BPS-L and OVA+BPS-H groups showed similar levels as compared with the OVA group. Compared with OVA alone, BPS with OVA decreased f, TVb, and MVb values (Figure 4b–d), and these changes were more remarkable in the OVA-BPS-M group.

### 2.3. BPS Elevates Allergen-Specific Ig Antibody Production in Serum

To confirm the adjuvant activity of BPS in allergic asthma, we examined serum OVA-IgE and IgG_1_ production 48 h after the last intratracheal administration. The OVA+BPS groups had higher OVA-IgE and IgG_1_ levels than the OVA group (Figure 5). In particular, OVA+BPS-M significantly increased serum OVA-IgE and IgG_1_ levels compared with OVA alone (*p* < 0.05). OVA-IgG_1_ was also significantly increased in the OVA+BPS-H group (*p* < 0.05), but its level was lower than that of the OVA+BPS-M group.

### 2.4. BPS Enhances Allergen-Induced Protein Production of Inflammatory Molecules in the Lung

To evaluate the protein levels of cytokines and chemokines in the lungs, we analyzed the lung homogenate supernatants by ELISA 48 h after the last intratracheal instillation. The levels of IL-5, IL-13, IL-33, and CCL11/Eotaxin significantly increased in the OVA+BPS-M and OVA+BPS-H groups compared with those in the OVA group, but greater in the OVA+BPS-M group (Figure 6). Meanwhile, OVA+BPS-L did not significantly change the expression of all molecules compared with OVA alone. No changes were observed in IL-4, RANTES, and IFN-γ expression in either group with or without OVA. No changes were observed in the group that only received BPS.

### 2.5. BPS Disrupts ER Expression in the Lung

To confirm whether BPS contributes to the exacerbation of allergic symptoms via ERs, we examined the mRNA levels of ERα, ERβ, and GPER in the lungs. The mRNA level of *Era* did not significantly change in all groups (Figure 7a,b). The OVA+BPS-L and OVA+BPS-H groups increased *Erb* expression compared with the OVA group; however, the OVA+BPS-M group revealed no significant effect (Figure 7c,d). Furthermore, *GPER* reduced dose-dependently in BPS with or without OVA (Figure 7e,f).

### 2.6. BPS Activates MLN Cells

To assess whether BPS promotes OVA-induced MLN cell activation, we examined the total cell count, cell surface marker expression, and cytokine secretion in OVA-restimulated MLN cells 48 h after the last intratracheal instillation. Total cell count and the percentage of MHC class II^+^ CD86^+^ cells in MLNs were greater in the OVA+BPS-M group than in the OVA group (*p* < 0.05, Figure 8a,b). The OVA+BPS-H group also showed a similar trend, but changes were not statistically significant. Conversely, the ratio of CD4 (TCRβ^+^CD4^+^ cells)/CD8 (TCRβ^+^CD8^+^ cells) did not change. The percentage of plasmacytoid dendritic cells (pDCs: CD11c^+^PDCA-1^+^) and conventional DCs (cDCs: CD11c^+^PDCA-1^-^) to total cells showed no changes (Figure 8c,d). However, by BPS exposure in allergic asthmatic mice, the percentage of cDC1 subsets (CD11c^+^PDCA-1^-^CD8a^+^CD11b^-^) to total cDC subsets decreased, and the percentage of cDC2 subsets (CD11c^+^PDCA-1^-^CD8a^-^CD11b^+^) to total cDC subsets increased (Figure 8e,f). These alterations were more prominent in the OVA+BPS-M group. Moreover, cell proliferation and cytokine production in the culture supernatant of MLN cells after OVA restimulation were determined in OVA-sensitized mice. The OVA+BPS-M and OVA+BPS-H groups had significantly greater cell proliferation than the OVA group (Figure 9a). The protein levels of IL-4, IL-5, and IL-13 also increased in the OVA+BPS-M and OVA+BPS-H groups compared with the levels in the OVA group (Figure 9b–d). The level of IFN-γ also showed a similar trend, but changes were not statistically significant (Figure 9e).

## 3. Discussion

This study investigated the effects of oral exposure to low doses of BPS (equivalent to doses of 0.04, 0.4, and 4 μg/kg/day) in allergic asthmatic mice. Compared with OVA alone, BPS-M and BPS-H with OVA enhanced allergic pulmonary inflammation, Th2 cytokine and chemokine production, and serum OVA-specific Ig secretion. In addition, MLN cells were activated following BPS exposure in OVA-sensitized mice. Notably, these results were more remarkable in the OVA+BPS-M group than in the OVA+BPS-H group. Although the association between BPA and allergic diseases such as allergic asthma has been extensively studied in animals, the effects of BPS remain poorly examined. This study is the first to show that oral exposure to low-dose BPS aggravates allergic asthma in an animal model.

Allergic asthma is characterized by the induction of type 2 immune responses with allergen-specific IgE production and eosinophil accumulation in the inflamed tissues [33]. We found that BPS elevates the protein expression levels of IL-5, IL-13, IL-33, and CCL11/Eotaxin in OVA-treated lungs. Conversely, Th1 cytokines such as IFN-γ demonstrated no significant changes. Th2 cytokines and chemokines play an important role in the pathophysiology of allergic diseases, including allergic asthma [34]. IL-5 promotes the growth, differentiation, activation, and survival of eosinophils and regulates eosinophil homing and migration into tissues [35,36]. IL-13 has several actions similar to IL-4, and it induces B-cell IgE class switch and mucus hypersecretion [37]. Mucus production and hypersecretion are important in asthma. IL-33, which is a member of the IL-1 family, plays critical roles in both innate and adaptive immunity. In innate immunity, IL-33 activates eosinophils such as type 2 innate lymphoid cells (ILC2s), basophils, and mast cells to initiate immune responses against invading pathogens or other environmental factors. In adaptive immunity, IL-33 regulates dendritic cell (DC) function and promotes Th2, Tfh, and Treg cell development [38]. In the present study, the exacerbation of allergic asthma may be caused by the activation of both innate and acquired immunity by BPS. CCL11/Eotaxin is an eosinophil-specific chemoattractant and a member of the C-C branch of the chemokine family. CCL11/Eotaxin elevates in various inflammatory diseases with eosinophilic infiltration, such as allergic asthma [39,40], allergic rhinitis [41,42], and atopic dermatitis [43,44]. In addition, CCL11/Eotaxin is a selective ligand of the C-C chemokine receptor 3 (CCR3), which is expressed on eosinophils, basophils, and Th2 lymphocytes. Furthermore, the OVA+BPS-M group had a remarkably increased mRNA level of CCR3 compared with the OVA group. Taken together, oral exposure to BPS may enhance eosinophilic infiltration and goblet cell hyperplasia by activating type 2 immunity resulting in exacerbated allergic pulmonary inflammation.

Bisphenols are well-known EDCs. Among them, BPA is a representative EDC that modifies hormone receptors, such as ERs, resulting in adverse effects. ERs consist of two classical receptors (ERα and ERβ) and a nonclassical receptor GPER, and they regulate the immune and endocrine systems [45,46]. In addition, estrogen plays inconsistent effects on allergic responses in the lung via ER regulation [47,48,49,50]. Our previous study revealed that BPA may enhance allergic asthma symptoms by activating ERβ in the lungs [51]. In addition, other EDCs such as organophosphorus flame retardants may aggravate allergic asthma via ER disruption. Tris(2-butoxyethyl) phosphate also increased the mRNA levels of *Era*, but not *Erb*, in allergic asthmatic mice [52]. Although tris(1,3-dichloro-2-propyl)phosphate decreased the mRNA levels of *GPER* dose-dependently in the lungs with or without OVA sensitization, ERα and ERβ showed no significant changes [53]. Thus, we examined gene expression in the lungs to confirm the contribution of ERs in allergic asthma exacerbation by BPS exposure. The mRNA level of ERα did not significantly change. ERβ increased in the OVA+BPS-L and OVA+BPS-H groups, but not the OVA+BPS-M group, compared with that in the OVA group. ERβ has an anti-inflammatory potential [54,55,56] and is reduced in the lung tissue cells of a mouse model of allergic airway inflammation [55]. BPS may disrupt allergic responses by alternating ERβ activation, but this effect may be different depending on the exposure dose of BPS. To clarify the involvement of ERs in the exacerbation of allergic asthma caused by BPS exposure, we need to investigate it using ER knockout mice or specific agonists/antagonists for ERs. Moreover, ERs are ubiquitously expressed in major organs and various cells, including immune cells; thus, BPS may exacerbate allergic asthma when multiple immune cells are modified. Further identifying the cells that contribute to the exacerbation of allergic asthma caused by BPS exposure is necessary.

We also found that GPER decreased depending on the BPS dose with or without OVA. In allergic asthmatic mice, GPER agonist attenuated airway hyperresponsiveness, inflammatory cell accumulation, and Th2 cytokine production (IL-5 and IL-13) in BAL fluid [57]. Accordingly, GPER reduction in the lungs caused by BPS exposure might suppress anti-inflammatory responses. An in vitro study showed that BPS induces ERK phosphorylation at the same level as estradiol (E2), which was attenuated by GPER inhibitor [58]. In human mature adipocytes and stromal vascular fraction cells, BPA enhances the accumulation of inflammatory molecules (MCP-1, IL-6, and IL-8) and induces the production of GPER, but not ERα and ERβ [59]. Thus, BPS may have the potential of generating both inflammatory and anti-inflammatory effects, and these responses may change depending on the exposure doses, exposure periods, and target organs. However, the mechanisms of allergic asthma exacerbation caused by BPS exposure require further research.

Next, we examined MLN cell activation by BPS exposure. BPS-M and BPS-H exposure in OVA-sensitized mice increased the total cell count and APC activation (MHC class II^+^CD86^+^ and cDC2 [CD11c^+^PDCA-1^-^CD8a^-^CD11b^+^] cells) but did not alter T cell composition. In addition, the OVA-restimulated cell proliferation and Th2 cytokine production (IL-4, IL-5, and IL-13) in the culture supernatant increased. These results were more remarkable in the OVA+BPS-M group.

DCs are the most potent APCs that act as a bridge between innate and adaptive responses. Allergen-activated DCs are essential for inducing Th-cell differentiation from naïve T cells in the MLN and accelerating pulmonary inflammation by allergen exposure [60]. DCs are divided into two distinct subsets according to their immunophenotype and functional properties: pDCs and cDCs [61,62,63]. pDCs are primary producers of type-I IFN for host defense against viral infection [64,65], whereas cDCs are important for innate immune responses as well as for initiating and regulating T cell responses [66]. cDCs are further divided into two major subsets: type 1 (cDC1) and type 2 (cDC2) [66,67]. The main features of cDC1 are the cross-presentation of antigens to CD8^+^ T cells and the direct presentation of antigens to CD4^+^ T cells, causing a Th1 response, whereas cDC2 presents extracellular antigens directly to CD4^+^ T cells and induces Th2 and Th17 immune responses [68,69]. In our previous study, intratracheal exposure to BPA enhanced APC activation in the MLN cells of OVA-induced allergic asthmatic mice [51]. Our in vitro study also showed that bone marrow-derived DCs differentiated by BPS exposure promotes OVA-specific antigen-presenting activity and Th2 cytokine production under IL-33 stimulation. Rank et al. reported that DCs promote naïve CD4^+^ T cell proliferation and differentiation and preferentially produce IL-5 and IL-13 in the presence of IL-33 [70]. In addition, the adoptive transfer of IL-33-treated DCs to naïve mice enhances lung airway inflammation. BPS may also enhance the activation of APCs (especially cDC2 subsets) and presentation of antigens to CD4^+^ T cells and induce cell proliferation and cytokine production by increasing IL-33 production in the lung. Therefore, BPS promotes Th2 cell migration from MLNs to the lung, leading to exacerbated allergic asthma.

In response to regulations in the EU, the US, China, and other countries regarding restrictions on BPA manufacture and utilization, BPA is gradually replaced by substitutes with similar chemical structures; these substitutes include BPS, bisphenol F, and BPAF. Zhang et al. reported that the estimated dietary exposure of BPS for adults is 22.2 ng/kg/day [71]. In another study, the estimated 24 h intake of BPS in pregnant women was 3.5 μg/kg/day (95th percentile: 0.12 μg/kg/day) [72]. Our current study showed that BPS at doses equivalent to 0.4 and 4 μg/kg/day promoted allergic asthma symptoms in mice. Therefore, low-dose exposure to BPS that is comparable to the level of actual human exposure may result in allergy exacerbation. The need for the regulatory action of “a group of bisphenols” as well as BPA has been discussed recently [73]. Understanding the potential risks of BPA alternatives, including BPS, is becoming increasingly important for maintaining human health; however, BPA alternative’s immunotoxicity remains insufficiently understood. Hence, our current study is important because it shows that BPS exposure has the potential to exacerbate allergic asthma.

## 4. Materials and Methods

### 4.1. Animals and Experimental Design

Four-week-old C3H/HeJSlc male mice were purchased from Japan SLC, Inc. (Shizuoka, Japan) to be used for the experiments. After 1 week of habituation, they were randomly divided into eight groups: (1) Vehicle, (2) low-dose BPS (BPS-L, 0.04 μg/kg/day), (3) moderate-dose BPS (BPS-M, 0.4 μg/kg/day), (4) high-dose BPS (BPS-H, 4 μg/kg/day), (5) ovalbumin (OVA), (6) OVA+BPS-L, (7) OVA+BPS-M, and (8) OVA+BPS-H. From 5 to 11 weeks of age, all mice in each group ingested BPS by drinking water at levels equivalent to 0, 0.04, 0.4, or 4 μg/kg/day of BPS. Exposure dose of BPS-H (4 μg/kg/day) is equivalent to the daily intake of 14 nmol/kg/day (approx. 3.5 μg/kg/day) estimated from urinary bisphenol concentrations in pregnant women in Canada [72]. We estimated the exposure amount of BPS according to the daily water consumption of approximately 3 mL in mice. Mice were under a chow diet based on soy-free AIN-76A to avoid phytoestrogenic effects (Oriental Yeast Co., Ltd., Tokyo, Japan). Food and water were provided *ad libitum*. Mice were housed in an animal facility maintained at 22 °C–26 °C and 40–69% humidity under a 12 h light/12 h dark cycle. To minimize background BPS exposure, we used polymethylpentene-made animal cages and polypropylene-made water bottles. Mice were then intratracheally instilled with 50 μL of aqueous solution every 2 weeks from 5 to 11 weeks of age under inhaled anesthesia with isoflurane (Pfizer Inc., New York, NY, USA). OVA-sensitized mice received 1 μg of OVA (20 μg/mL; Sigma-Aldrich Co., St. Louis, MO, USA) dissolved in phosphate-buffered saline (PBS; pH7.4; Thermo Fisher Scientific, Inc., Waltham, MA, USA), whereas nonsensitized mice received PBS. Furthermore, 48 h after the final instillation, the mice, which were already 11 weeks old, were euthanized through an intraperitoneal injection of sodium pentobarbital (150 mg/kg). The Animal Care and Use Committee of National Institute for Environmental Studies approved all the procedures, which conformed to the guidelines of the Care and Use of Laboratory Animals of the National Institute for Environmental Studies. Animals were humanely treated and were alleviated of suffering.

### 4.2. Retrieval of Bronchoalveolar Lavage (BAL) Fluid and Counting of Cell Number in Lavage Fluid

After cannulation via the trachea (5–6 mice/group), BAL fluids were collected by aspirating three times with 0.8 mL of preheated physiological saline (37 °C). The average recovery volume of BAL fluid was 90% of the volume instilled. Thereafter, the BAL fluid was centrifuged at 300× *g* for 10 min at 4 °C to collect its cells. The supernatants were removed and suspended in 1 mL of PBS. We stained all suspensions with Turk’s solution (Nacalai Tesque Inc., Kyoto, Japan) and then measured the total cell count using a hemocytometer. Differential cell counts were determined on cytospin-prepared slides (Sakura Seiki Co., Tokyo, Japan) using Diff-Quik staining (International Reagents Co., Kobe, Japan). Overall, 500 cells were counted using a light microscope (AX80; Olympus, Tokyo, Japan).

### 4.3. Preservation of Lung Tissue

After BAL, lungs (5–6 animals/group) were removed from the body cavity and immediately minced on ice. A part of the lung tissues was soaked in RNA later (QIAGEN N.V., Venlo, The Netherlands) and then stored at −20 °C for gene expression analysis; the rest was frozen in liquid nitrogen and then stored at −80 °C for protein expression analysis.

### 4.4. Evaluation of Pulmonary Function

Airway responsiveness to methacholine chloride solution (MCh) (Sigma-Aldrich, St. Louis, MO, USA) was measured 24 h after the last OVA intratracheal instillation, using noninvasive whole-body plethysmography (WBP) (Buxco FinePointe System, Buxco, Wilmington, USA) according to the manufacturer’s instructions (6 animals/group). The mice were acclimated for 5 min and then nebulized with 50 μL of PBS as baseline, followed by 50 μL each of a twofold dilution series of MCh solution (3.125, 6.25, 12.5, 25, and 50 mg/mL in PBS) sequentially. After nebulization, airway responsiveness to MCh was monitored for 3 min and assessed for the rate of increase from the baseline of the enhanced pause (Penh), tidal volume (TVb), minute volume (MVb), and respiratory rate (f).

### 4.5. Histopathological Evaluation of the Lungs

The lungs were fixed in 10% neutral buffered formalin 48 h after the final intratracheal instillation. We then embedded tissue sections in paraffin and prepared 4 μm-thick slices (four animals/group). The histological specimens were stained with hematoxylin and eosin (H&E) and also periodic acid–Schiff (PAS) to evaluate eosinophil and lymphocyte infiltration in the airway and identify goblet cell proliferation in the bronchial epithelium, respectively. Subsequently, histological findings were assessed using an Olympus BX43 microscope. The degree of eosinophil and lymphocyte infiltration in the airways or goblet cell proliferation in the bronchial epithelium was graded in a blind fashion as follows: 0 = not present, 0.5 = slight, 1 = mild, 1.5 = mild to moderate, 2 = moderate, 2.5 = moderate to marked, and 3 = marked. The ratings of 0.5 (lowest rating), 1, 1.5, 2, 2.5, and 3 indicated an inflammatory reaction affecting <10%, 10–20%, 20–30%, 30–40%, 40–50%, and >50%, respectively, of the airways or goblet cells.

### 4.6. Quantification of Protein Levels in Lung Tissues and Antigen-Specific Immunoglobulin in Serum

Lung tissues were homogenized in 10 mM potassium phosphate buffer (pH 7.4) containing 0.1 mM ethylenediaminetetraacetic acid (Sigma-Aldrich), 0.1 mM phenylmethanesulfonyl fluoride (Nacalai Tesque), 1 μM pepstatin (Peptide Institute, Inc., Osaka, Japan), and 2 μM leupeptin (Peptide Institute). Thereafter, the homogenates were centrifuged at 105,000× *g* for 1 h at 4 °C, and the supernatants were dispensed into aliquots and stored at −80 °C until use. Cytokines and chemokines (interleukin (IL)-5, IL-13, IL-33, CCL11/Eotaxin, and regulated on activation, normal T cell expressed and secreted (RANTES)) in lung homogenate assessed by Luminex^®^ Assay were obtained from R&D Systems (Minneapolis, MN, USA). In addition, total protein count was measured using Bradford ULTRA^™^ (Novexin Ltd., Cambridge, England). The abundance of the target protein was normalized to the total amount of protein.

Following euthanasia, the chest and abdominal walls were opened, and blood was extracted via cardiac puncture. By centrifugation at 3000× *g* for 10 min at 20 °C (5–6 animals/group), serum was recovered using the MiniCollect^®^ Tube (450534, Greiner Bio-One GmbH, Kremsmünster, Austria) and was stored at −80 °C until use. Furthermore, OVA-specific IgE and IgG_1_ were measured using mouse anti-OVA-IgE enzyme-linked immunosorbent assay (ELISA) Kit (Shibayagi Co., Gunma, Japan) and anti-OVA-IgG_1_ ELISA Kit (Shibayagi Co.) according to the manufacturer’s instructions.

### 4.7. Real-Time Reverse Transcription Polymerase Chain Reaction (RT-PCR) Analysis

Total RNA from lungs was extracted using RNeasy mini kit (QIAGEN) according to the manufacturer’s instructions (6 animals/group). We used the NanoDrop spectrometer (Thermo Fisher Scientific) to assess the total RNA concentration spectrophotometrically, and High-Capacity RNA-to-cDNA™ Kit (Thermo Fisher Scientific) to reverse-transcribe it to cDNA. The mRNA expression levels of estrogen receptor alpha (*Era*), estrogen receptor beta (*Erb*), and G protein-coupled estrogen receptor 1 (*GPER*) were measured using the StepOne Plus™ Real-time PCR System (Thermo Fisher Scientific). Next, RT-PCR was conducted at 50 °C for 2 min, 95 °C for 10 min, 95 °C for 15 s, and 60 °C for 1 min, with the last two steps repeated for 40 cycles. Data were then analyzed by the critical threshold (ΔCT) and the comparative critical threshold (ΔΔCT) methods using StepOne Plus™ Software version 2.2.2. The relative intensity was normalized to an endogenous control gene (hypoxanthine phosphoribosyltransferase 1, *Hprt1*). TaqMan probes and pairs for target genes were designed and purchased from Thermo Fisher Scientific; however, these sequences were not disclosed.

### 4.8. Preparation of MLN Cells and Flow Cytometry Analysis

To prepare single-cell suspensions from MLNs (6 animals/group), we strained the cells through a sterile stainless wire mesh into PBS (−) (pH 7.4; Takara Bio Inc., Shiga, Japan). MLN cells were collected by centrifugation at 400× *g* for 5 min at 20 °C, followed by red blood cell lysis with ammonium chloride. After washing with PBS (−), we resuspended the cells in R10 culture medium comprising the Gibco RPMI 1640 medium (Thermo Fisher Scientific) supplemented with 10% heat-inactivated fetal bovine serum (MP Biomedicals Inc., Eschwege, Germany), penicillin (100 U/mL), streptomycin (100 μg/mL; Sigma-Aldrich), and 2-mercaptoethanol (50 μM; Thermo Fisher Scientific). The total cell count was recovered, and cell viability was determined using trypan blue exclusion (Thermo Fisher Scientific).

Moreover, fluorescence of live MLN cells was measured using the DxFLEX flow cytometer (Beckman Coulter Inc., Pasadena, CA, USA). We used the following anti-mouse REAfinity antibodies (recombinant human IgG1; Miltenyi Biotec GmbH, Bergisch Gladbach, Germany): anti-PDCA-1 (REA818, fluorescein isothiocyanate (FITC)-conjugated), anti-CD11b (REA592, phycoerythrin (PE)-conjugated), anti-CD8a (REA601, PerCP-conjugated), anti-CD11c (REA754, antigen-presenting cell (APC)-conjugated), anti-MHC class II I-A/I-E (REA813, FITC-conjugated), anti-CD86 (REA1190, PE-conjugated), anti-TCR β (REA318, FITC-conjugated), anti-CD4 (REA604, PE-conjugated), and isotype control (REA293) for each conjugate. We prepared the cell samples according to the manufacturers’ instructions and incubated them with antibodies for 30 min on ice. All data were analyzed using the FlowJo software (BD Biosciences, San Jose, CA, USA).

### 4.9. Proliferation and Cytokine Production of MLN Cells

MLN cells (1 × 10^6^/mL) were cultured with or without OVA (100 μg/mL) in 200 μL of R10 medium in 96-well flat-bottom plates. These cultures were performed thrice at 37 °C in a 5% CO_2_/95% air atmosphere. After 91 h, culture supernatant was collected and stored at −80 °C until use. Next, we added 5-bromo-2′-deoxyuridine to each well 20 h before cell proliferation measurement using an ELISA kit (Roche Molecular Biochemicals, Mannheim, Germany) according to the manufacturer’s instructions. We also used the ELISA kits to measure the IL-4, IL-5, IL-13, interferon-gamma (IFN-γ) levels (Thermo Fisher Scientific) in the cell culture supernatants.

### 4.10. Statistical Analysis

All data were statistically analyzed using BellCurve for Excel statistical software (Social Survey Research Information Co., Ltd., Tokyo, Japan). Statistical significance was determined by one-way analysis of variance and then the Tukey or Scheffe test. Nonparametric analysis was conducted using the Kruskal–Wallis test, followed by Steel’s test. A *p* value less than 0.05 was considered statistically significant.

## 5. Conclusions

Oral exposure to low-dose BPS can aggravate allergic responses by enhancing type 2 immune responses in the lung and alternating the lymph node microenvironments. Disruption of ER expression in the lung may have caused these adverse effects.

## Figures and Tables

**Figure 1 ijms-23-10790-f001:**
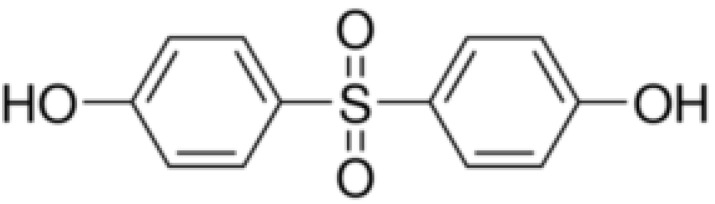
Chemical structure of bisphenol S.

**Figure 2 ijms-23-10790-f002:**
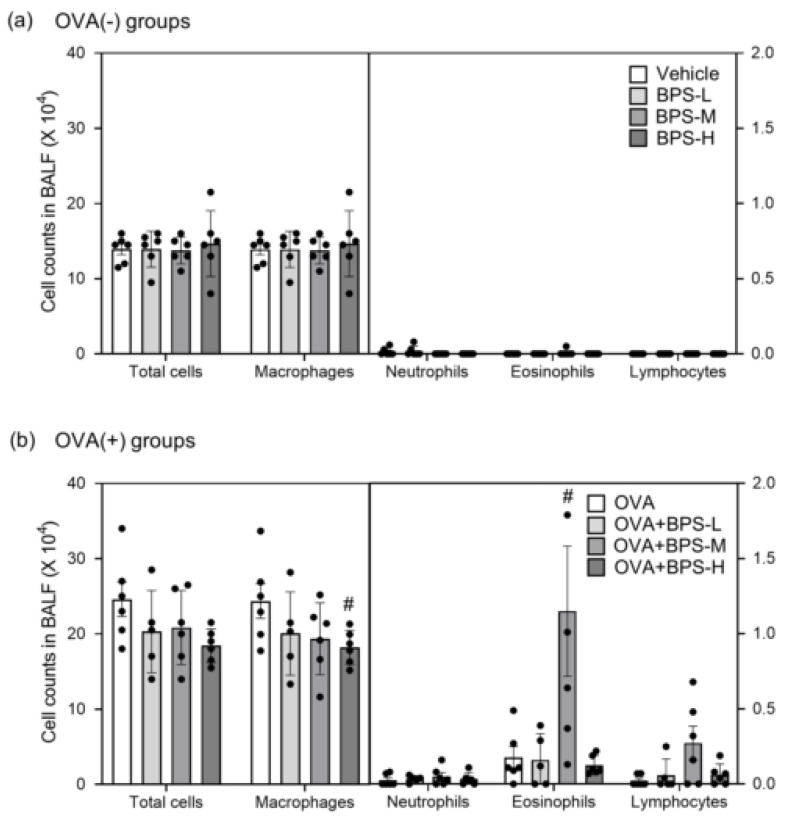
Cellular profile in BAL fluid. (**a**) Vehicle-administered groups. (**b**) OVA-administered groups. Data are expressed as mean ± SE for 5–6 animals per group. # *p* < 0.05 vs. OVA group. BAL, bronchoalveolar lavage; OVA, ovalbumin; SE, standard error.

**Figure 3 ijms-23-10790-f003:**
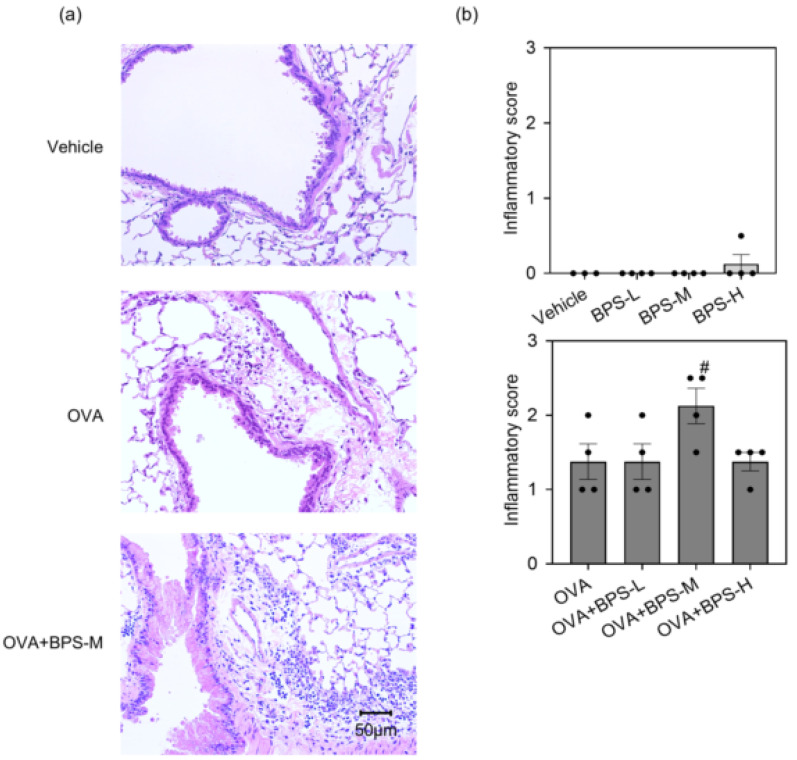
Histopathological findings in the lung. Histopathological changes were evaluated using hematoxylin and eosin staining 48 h after the final intratracheal administration. (**a**) Histopathological images. (**b**) Inflammatory score. Data are expressed as mean ± SE for 3–4 animals per group. # *p* < 0.05 vs. OVA group. OVA, ovalbumin; SE, standard error.

**Figure 4 ijms-23-10790-f004:**
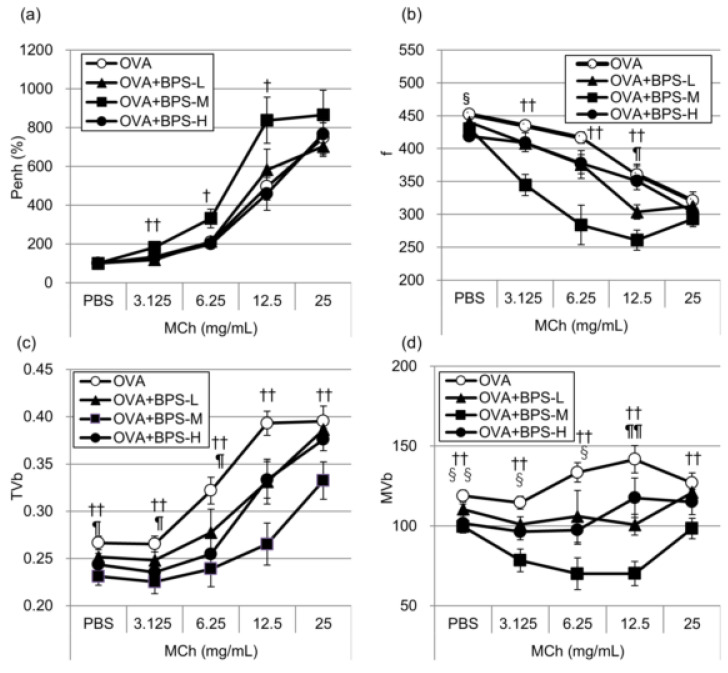
Changes in airway responsiveness in OVA-sensitized mice. Airway responsiveness to methacholine chloride 24 h after the final OVA intratracheal administration was measured using whole-body plethysmography. (**a**) Enhanced pause values. (**b**) Respiratory rate (f). (**c**) Tidal volume. (**d**) Minute volume. Data are expressed as mean ± SE for 5–6 animals per group. ¶ *p* < 0.05, OVA+BPS+L group vs. OVA group; ¶¶ *p* < 0.05, OVA+BPS+L group vs. OVA group; † *p* < 0.05, OVA+BPS+M group vs. OVA group; †† *p* < 0.01, OVA+BPS+M group vs. OVA group; § *p* < 0.05 OVA+BPS+H group vs. OVA group; §§ *p* < 0.01 OVA+BPS+H group vs. OVA group. BPS, bisphenol S; OVA, ovalbumin; SE, standard error.

**Figure 5 ijms-23-10790-f005:**
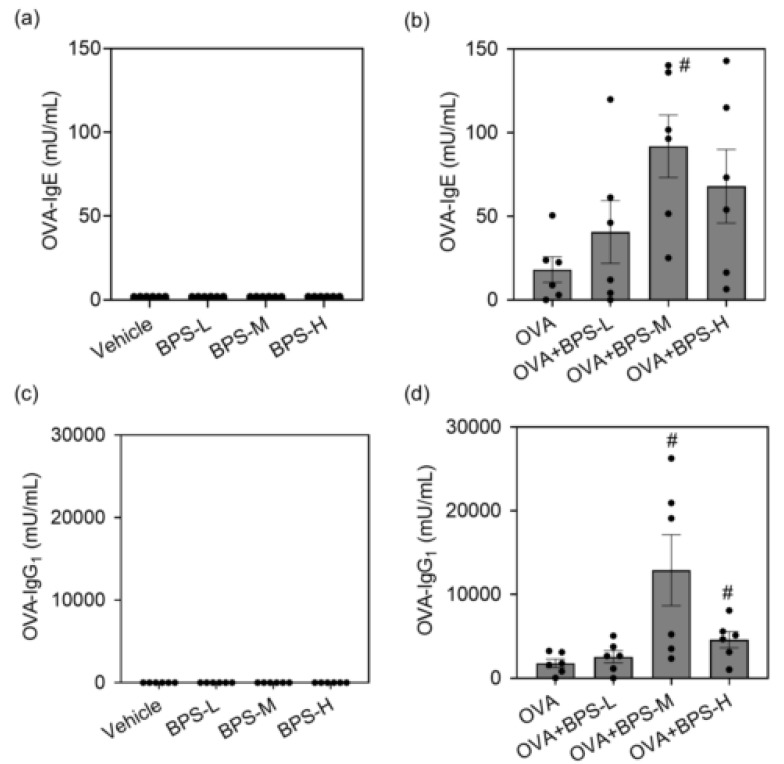
Serum levels of OVA-specific immunoglobulin (Ig) antibodies. OVA-specific IgE and IgG_1_ were measured 48 h after the last intratracheal administration by ELISA. (**a**,**b**) OVA-IgE. (**c**,**d**) OVA-IgG_1_. Data are expressed as mean ± SE for 5–6 animals per group. # *p* < 0.05 vs. OVA group. ELISA, enzyme-linked immunosorbent assay; OVA, ovalbumin; SE, standard error.

**Figure 6 ijms-23-10790-f006:**
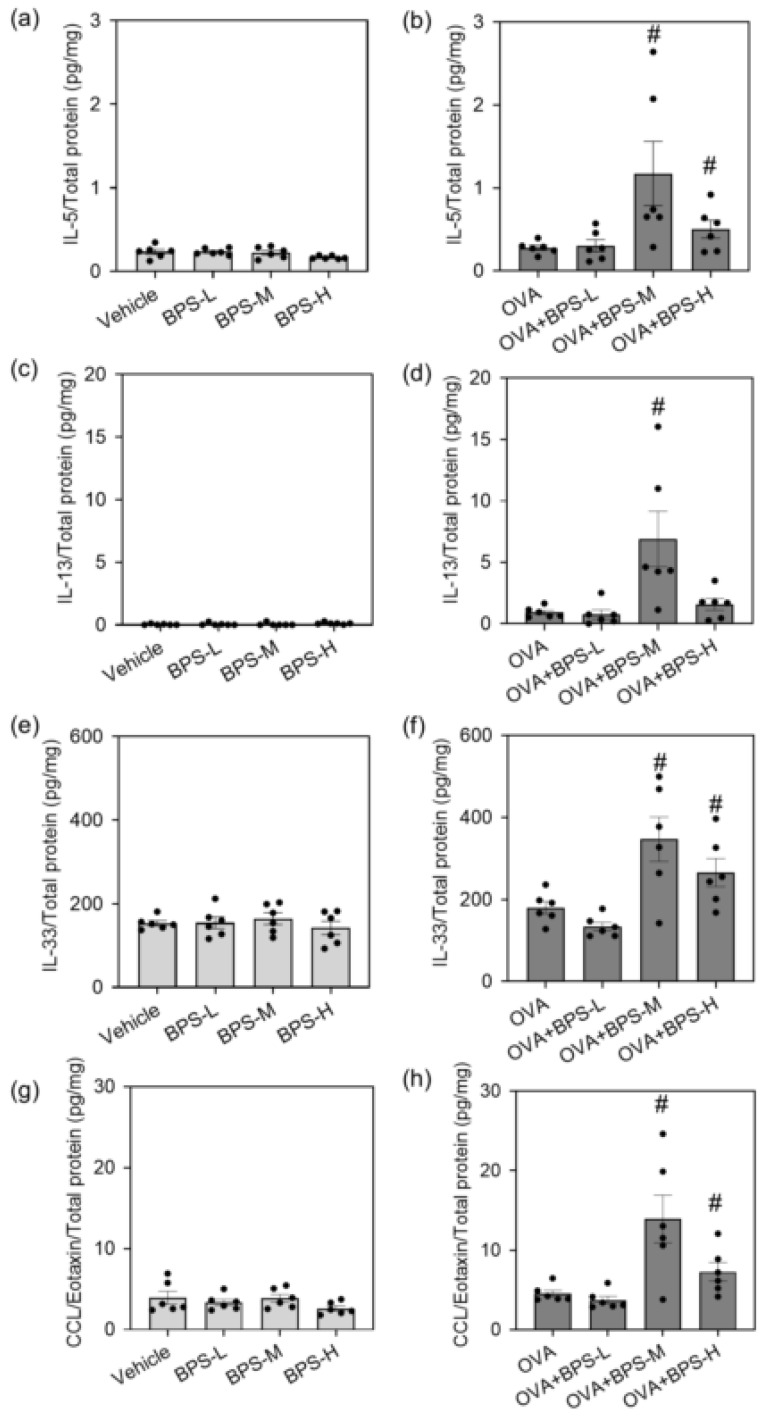
Protein levels of cytokines and chemokines in the lung. Protein levels in the lung homogenate supernatants were measured 48 h after the last intratracheal administration by ELISA. (**a**,**b**) IL-5. (**c**,**d**) IL-13. (**e**,**f**) IL-33. (**g**,**h**) CCL11/Eotaxin. The abundance of the target protein was normalized to the total amount of protein. Data are expressed as mean ± SE for 5–6 animals per group. # *p* < 0.05 vs. OVA group. ELISA, enzyme-linked immunosorbent assay; IL, interleukin; OVA, ovalbumin; SE, standard error.

**Figure 7 ijms-23-10790-f007:**
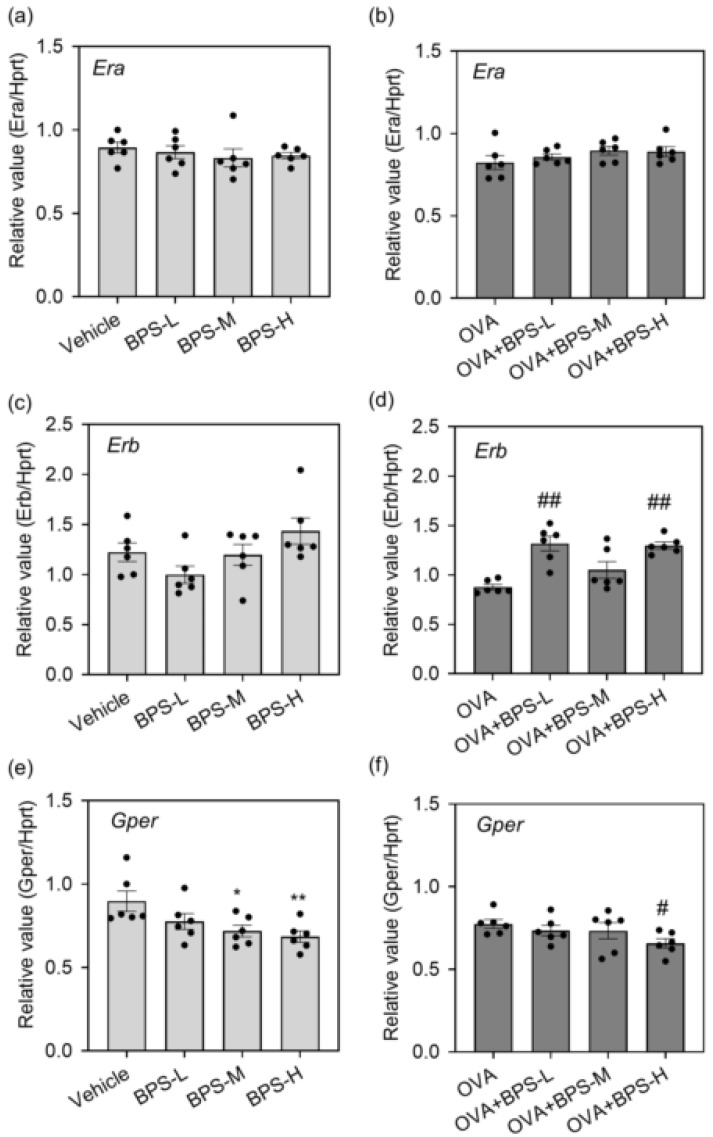
Gene expression of estrogen receptors in the lung. mRNA levels were analyzed 48 h after the last intratracheal administration by reverse transcription polymerase chain reaction. (**a**,**b**) *Era*. (**c**,**d**) *Erb*. (**e**,**f**) *Gper*. Data are expressed as mean ± SE for 5–6 animals per group. The relative intensity was normalized to housekeeping gene (*Hprt1*). * *p* < 0.05 vs. Vehicle group, ** *p* < 0.01 vs. Vehicle group, # *p* < 0.05 vs. OVA group, ## *p* < 0.01 vs. OVA group. OVA, ovalbumin; SE, standard error.

**Figure 8 ijms-23-10790-f008:**
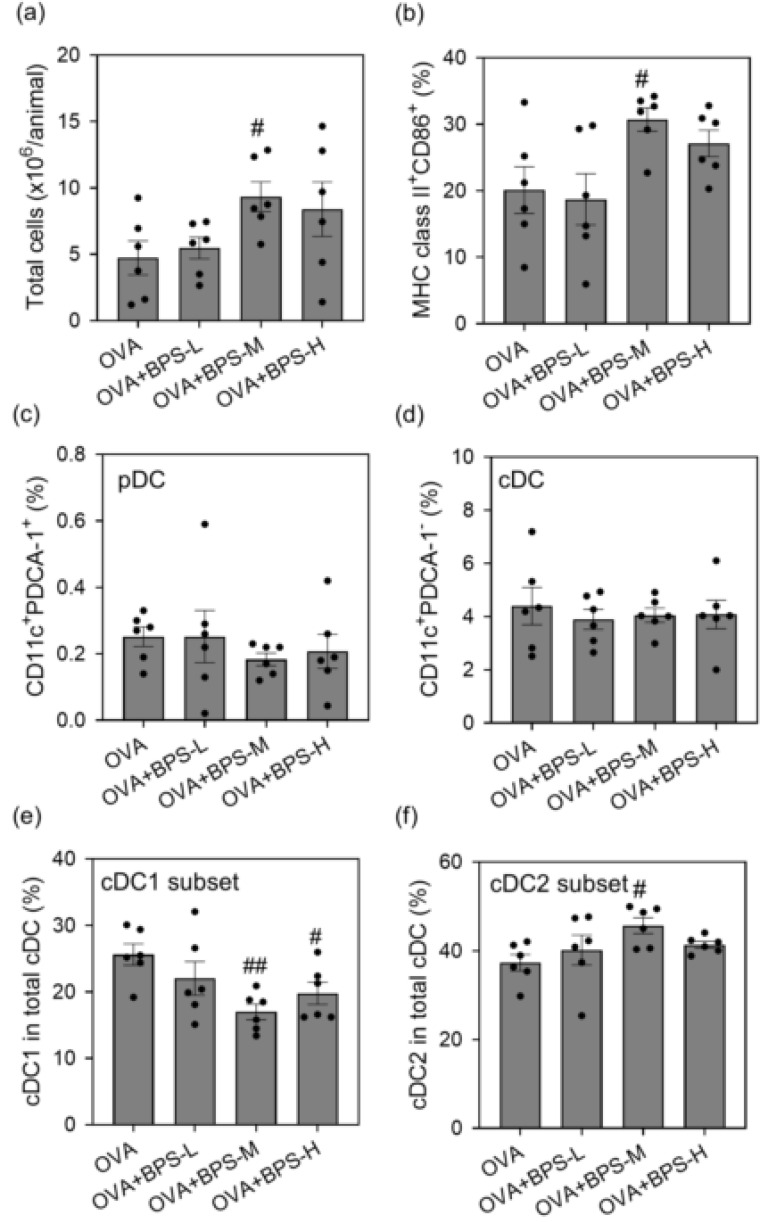
Expression of cell number and cell surface molecules in MLNs. Cell surface molecule expression was determined via fluorescence-activated cell sorting analysis 48 h after the final OVA intratracheal administration. (**a**) Total cell number. (**b**) Percentage of MHC class II^+^ CD86^+^ cells. (**c**) Percentage of pDC. (**d**) Percentage of cDC. (**e**) Percentage of cDC1 in total cDC. (**f**) Percentage of cDC2 in total cDC. Data are expressed as means ± SE for 6 animals per group. # *p* < 0.05 vs. OVA group, ## *p* < 0.01 vs. OVA group. cDC, conventional dendritic cells; MLN, mediastinal lymph node; OVA, ovalbumin; pDC, plasmacytoid dendritic cells; SE, standard error.

**Figure 9 ijms-23-10790-f009:**
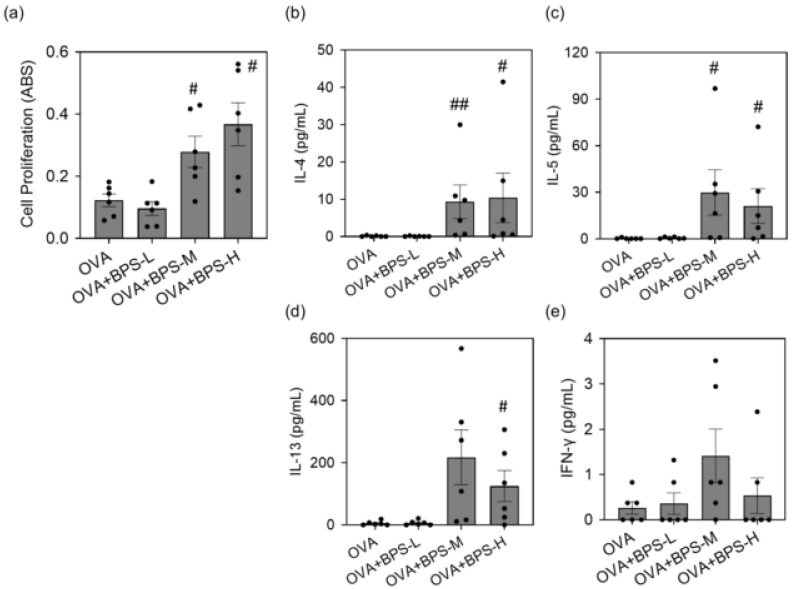
Activation of MLN cells. MLN cell proliferation and cytokine expression in the culture supernatant were analyzed 91 h after OVA restimulation. (**a**) Cell proliferation (Abs). (**b**) IL-4. (**c**) IL-5. (**d**) IL-13. (**e**) IFN-γ. Data are expressed as mean ± SE of 6 animals per group. # *p* < 0.05 vs. OVA group, ## *p* < 0.01 vs. OVA group. IFN, interferon; IL, interleukin; MLN, mediastinal lymph node; OVA, ovalbumin; SE, standard error.

## Data Availability

All data generated or analyzed during this study are available from the corresponding author on reasonable request.

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
