# Peer review of "Effects of Oral Exposure to Low-Dose Bisphenol S on Allergic Asthma in Mice"

_ijms, 2022, doi:10.3390/ijms231810790_

Round 1

Reviewer 1 Report

The authors prepared a manuscript investigating the consequences of oral-exposure of different doses of BPS for development of allergic airway inflammation. The study is of importance and interest for the general population since it uncovers the potential connection between low-dose BPS exposure and allergic asthma exacerbation.

The Manuscript has several major flaws.

11.)    The main concern arising from reviewing the manuscript is the observed lack of allergic stimulation. For example in Fig. 1 there are little changes in total cell counts between OVA- and OVA+ group (17x104 vs 24x104).

A successfully conducted OVA stimulation yields millions of cells, of those predominantly eosinophils. Generally, composition of BAL is better assessed with FACS, which is the method, the authors used for MLN composition determination. First of all, the authors should show that their model works by comparing the number of cells (and cell subtypes) between OVA- and OVA+. Alveolar macrophage numbers should serve as control and should stay constant between OVA- and OVA+. The authors should alternatively show primary data of microscopic slides and changes between groups. Moreover, successful OVA stimulation always leads to detectable increases in CCL11 and IL-4- which is actually the hallmark of Th2 inflammation. This is not observed in this manuscript (Fig6).

22.)    Why did the authors use C3H/HeJSlc mice, which are deficient in TLR4? Balb/c are more commonly used and can mount a better Th2 response. Additionally, the authors to not rationalize the use of male-only mice? Could female (potentially ovariectomized) mice be used?

33.)    Which cell type do the authors think is responsible for the observed effect on estrogen receptors? The assay was performed in total lung homogenates.

44.)    In order to fully prove the involvement of ER in the pathology of asthma exacerbation by BPS, the authors should repeat their model in an ER- knockout mouse. Alternatively, they could rescue their effect by using specific pharmacological agents (agonists/antagonists) of the receptors.

Minor: Scatter dot plot should be used instead of column data in all figures. This way, individual samples and their distribution would be recognizable.

Did the authors perform only one independent experiment with 5-6 mice per group? Or were more independent experiments conducted? If yes, how many?

Two-Way ANOVA should be used to analyze the results of Fig4., this test is not listed in methods.

Author Response

Thank you for your kind and appropriate comments. We have made revisions as ‘Track changes’ to your suggestions as follows:

Point 1: The main concern arising from reviewing the manuscript is the observed lack of allergic stimulation. For example in Fig. 1 there are little changes in total cell counts between OVA- and OVA+ group (17x104 vs 24x104).

A successfully conducted OVA stimulation yields millions of cells, of those predominantly eosinophils. Generally, composition of BAL is better assessed with FACS, which is the method, the authors used for MLN composition determination. First of all, the authors should show that their model works by comparing the number of cells (and cell subtypes) between OVA- and OVA+. Alveolar macrophage numbers should serve as control and should stay constant between OVA- and OVA+. The authors should alternatively show primary data of microscopic slides and changes between groups. Moreover, successful OVA stimulation always leads to detectable increases in CCL11 and IL-4- which is actually the hallmark of Th2 inflammation. This is not observed in this manuscript (Fig6).

Response: Thank you for your insightful comments. As you pointed out, in the pathogenesis of allergic asthma, inflammatory symptoms in general allergic asthmatic models appear quite strongly, including an increase in eosinophils, CCL11, and IL-4 (e.g., allergen + adjuvant for immunization and then OVA inhalation for challenge). However, in our asthmatic mouse model, the inflammatory pathology is comparably mild to detect the effects of environmental chemicals; thus, the degree of increase in cell number and lung inflammation is not too remarkable. However, the administration of OVA caused a slight increase in the total cell counts and enhancement of OVA-specific antibody production and eosinophilic infiltration. Therefore, allergic sensitization is established. We would like to confirm the cellular composition of BAL by FACS in future studies.

Point 2: Why did the authors use C3H/HeJSlc mice, which are deficient in TLR4? Balb/c are more commonly used and can mount a better Th2 response. Additionally, the authors to not rationalize the use of male-only mice? Could female (potentially ovariectomized) mice be used?

Response: Thank you for your comments. As you pointed out, BALB/c mice are often used as a model for allergic asthma. C3H/HeJSlc mice are highly sensitive to environmental pollutants (e.g., chemicals and air pollutants) as well as allergens; thus, we used C3H/HeJSlc for our study. In addition, we chose C3H/HeJSlc to reduce TLR4-mediated responses because this study does not focus on TLR4.

We selected males because females are known to have greater data variability than males because of the estrus cycle (including the ER effects). However, the difference in the effects of sex variation is a very important point that we would like to investigate in the future.

Point 3: Which cell type do the authors think is responsible for the observed effect on estrogen receptors? The assay was performed in total lung homogenates.

Response: Thank you for your important remarks. ERs are expressed in various cells and tissues. In particular, ERα and ERβ are expressed in B-cell, NK cell, and bone marrow-derived DC (10.1016/j.cellimm.2015.01.018). ERα is also expressed in CD4+ T cell, plasmacytoid DC, and splenic DC. In addition, GPER is expressed in peripheral B and T cells, monocytes, eosinophils, and neutrophils (10.3389/fendo.2020.579420). Moreover, BPS exposure may activate several immune cells resulting in allergic asthma regulation. The related sentences were revised in Discussion, as follows: “ERs are ubiquitously expressed in major organs and various cells, including immune cells; thus, BPS may exacerbate allergic asthma when multiple immune cells are modified. Further identifying the cells that contribute to the exacerbation of allergic asthma due to BPS exposure is necessary.” (P17, L2–6)

Point 4: In order to fully prove the involvement of ER in the pathology of asthma exacerbation by BPS, the authors should repeat their model in an ER- knockout mouse. Alternatively, they could rescue their effect by using specific pharmacological agents (agonists/antagonists) of the receptors.

Response: We agree that examination using ER KO mice or agonists/antagonists of ER is necessary to clarify the involvement of ERs. We would like to consider this in the future. We have revised the related sentence in Discussion, as follows: “To clarify the involvement of ERs in the exacerbation of allergic asthma caused by BPS exposure, we need to investigate it using ER knockout mice or specific agonists/antagonists for ERs.” (P16, L25–P17, L2)

Minor: Scatter dot plot should be used instead of column data in all figures. This way, individual samples and their distribution would be recognizable.

Response: We revised all figures, except Figure 4, because the graph in Figure 4 becomes complex when displayed in dots.

Did the authors perform only one independent experiment with 5-6 mice per group? Or were more independent experiments conducted? If yes, how many?

Response: We conducted two experiments for OVA administration and one experiment for vehicle administration. For the OVA-treated group, the results of each experiment are shown, and both experiments showed similar trends.

Two-Way ANOVA should be used to analyze the results of Fig4., this test is not listed in methods.

Response: Figure 4 illustrates the comparison between the OVA+BPS and OVA groups at each concentration of methacholine; thus, we analyzed the results using one-way ANOVA, as with the other data.

Reviewer 2 Report

The manuscript by Rie Yanagisawa and co-authors describes that exposure of OVA allergic mice to low-dose bisphenol S seems able to aggravate allergic responses in the lungs and the MLNs of these mice. The study is well conducted and presented; some minor points should be addressed by the authors.

Minor comments:

1)    Page 2, material and methods, chapter 2.1: please include the reference number in the last sentence - Liu, 2018

2)    How many mice per group were analysed for histologic evaluation of the lungs?

3)    Page 5, results, figure 3: the group OVA+BPS-L is not shown; would it be possible to do a grading on a semi-quantitative scale of histological HE stained lung slides (e.g. grade 0 – 3 with increasing inflammatory values)? At first glance, there seems to be no difference between HE stained lung sections from group OVA and group OVA+BPS-M.

4)    Page 9, results, chapter 3.4, figure 6: do the same trends show in the Luminex Assay data without normalization to total protein content?

Author Response

Thank you for your kind and appropriate comments. We have made revisions as ‘Track changes’ to your suggestions as follows:

Point 1: Page 2, material and methods, chapter 2.1: please include the reference number in the last sentence - Liu, 2018

    Response: Thank you for addressing this point. We have revised the reference number.

Point 2: How many mice per group were analysed for histologic evaluation of the lungs?

Response: We added the number of animals to the legend of Figure 3 as follows: “(a) Histopathological images. (b) Inflammatory score. Data are expressed as mean ± SE for four animals per group. # P < 0.05 vs. OVA group.” (P35, L11-13)

Point 3:  Page 5, results, figure 3: the group OVA+BPS-L is not shown; would it be possible to do a grading on a semi-quantitative scale of histological HE stained lung slides (e.g. grade 0 – 3 with increasing inflammatory values)? At first glance, there seems to be no difference between HE stained lung sections from group OVA and group OVA+BPS-M.

Response: In line with your comment, we added inflammatory score data in the OVA groups and replaced the histological image of OVA group in Figure 3. Unfortunately, semi-quantified results showed no difference in goblet cell hyperplasia, so we revised the related description, as follows: “In pulmonary inflammation evaluation using H&E (Figure 3a) and PAS staining, groups without OVA sensitization showed no changes. OVA enhanced eosinophil and lymphocyte accumulation in the peribronchial and perivascular regions and goblet cell hyperplasia in the bronchial epithelium. The accumulation of eosinophils and lymphocytes was more remarkable in the OVA+BPS-M group than in the OVA group (P < 0.05, Figure 3b). In the OVA+BPS-H group, pulmonary inflammation in the lungs were rather attenuated. However, goblet cell hyperplasia did not significantly change in either group with OVA (data not shown).” (P11, L13-21)

Point 4:  Page 9, results, chapter 3.4, figure 6: do the same trends show in the Luminex Assay data without normalization to total protein content?

Response: Thank you for your comments. Given that the inflammatory responses became more intense, the levels of target proteins were slightly counteracted by an increase in the total lung protein. Nevertheless, a similar trend was observed.

Round 2

Reviewer 1 Report

I have no further comments, concerns.